# Measuring Foot Abduction Brace Wear Time Using a Single 3-Axis Accelerometer [note 1]

**DOI:** 10.3390/s22072433

**Published:** 2022-03-22

**Authors:** Benjamin Griffiths, Natan Silver, Malcolm H. Granat, Ehud Lebel

**Affiliations:** 1School of Health and Society, University of Salford, Salford M5 4WT, UK; b.n.griffiths@salford.ac.uk; 2Shaare Zedek Medical Center, Jerusalem 9103102, Israel; doctornatan@gmail.com (N.S.); lebel@szmc.org.il (E.L.)

**Keywords:** foot abduction brace, clubfoot, compliance, wear, non-wear, infants, open access

## Abstract

The recommended treatment for idiopathic congenital clubfoot deformity involves a series of weekly castings, surgery, and a period of bracing using a foot abduction brace (FAB). Depending on the age of the child, the orthotic should be worn for periods that reduce in duration as the child develops. Compliance is vital to achieve optimal functional outcomes and reduce the likelihood of reoccurrence, deformity, or the need for future surgery. However, compliance is typically monitored by self-reporting, which is time-consuming to implement and lacks accuracy. This study presents a novel method for objectively monitoring FAB wear using a single 3-axis accelerometer. Eleven families mounted an accelerometer on their infant’s FAB for up to seven days. Parents were also given a physical diary that was used to record the daily application and removal of the orthotic in line with their treatment. Both methods produced very similar measurements of wear that visually aligned with the movement measured by the accelerometer. Bland Altman plots showed a −0.55-h bias in the diary measurements and the limits of agreement ranging from −2.96 h to 1.96 h. Furthermore, the Cohens Kappa coefficient for the entire dataset was 0.88, showing a very high level of agreement. The method provides an advantage over existing objective monitoring solutions as it can be easily applied to existing FABs, preventing the need for bespoke monitoring devices. The novel method can facilitate increased research into FAB compliance and help enable FAB monitoring in clinical practice.

## 1. Introduction

Idiopathic clubfoot is a congenital deformity that can be severely disabling and has a significant impact on mobility and quality of life if left untreated [1]. Treatment for idiopathic clubfoot is the Ponseti method, which involves a series of weekly castings, a Tendo-Achilles tenotomy in most cases, and a subsequent period of bracing using a foot abduction brace (FAB). Depending on the age of the child, the orthotic should be worn for predetermined periods that reduce in duration as the child develops [2]. Children treated appropriately are expected to have normal foot function, with no resulting disability [3]. However, for this treatment to be effective, compliance with wear-time protocols is vital. Failure to do so has been shown to increase the likelihood of reoccurrence and the need for future surgery [4,5,6].

Traditional methods of measuring compliance with FAB wear protocols involve the use of self-reporting measures and wear diaries [7]. Although these methods are simple to administer and easy to use by participants, questions regarding the validity of the data are common due to social bias when reporting, difficulty interpreting the questions and the inability to recall information accurately [8,9,10,11]. Clinical guidance on FAB wear protocols is driven by research using these methods [12], and it is important that these data are accurate and valid. Furthermore, these data must reflect the true wear time of the device to ensure that each patient’s treatment can be accurately assessed. An objective method of monitoring compliance with FAB wear-time protocols is vital to overcome these limitations and improve our understanding of FAB compliance.

More recently, sensor systems have been used to monitor FAB wear and overcome the limitations of subjective compliance measures. Morgenstein et al. [11] explored the use of pressure sensors housed within FAB binding insoles to determine wear duration per day and identify differences between reported and actual wear rates. The study found that actual wear rates for months 1, 2, and 3 of treatment were 91.7%, 86.8%, and 77.1%, while the self-reported wear rate were 94.9%, 95.6%, and 94.8%. This showed a clear difference between the objective and subjective measurements. Although these results provide valuable insight into the difference between reported and measured wear rates, there is no information on the validation of the devices’ measurements. Furthermore, the study had a high dropout rate attributed to the sensing device’s size and weight, highlighting the need for a more discrete and compact system.

Similarly, Richards et al. [13] and Sangiorgio et al. [14] investigated the use of FAB devices with integrated temperature sensors but did not present any form of data validation. Aroojis et al. [15] investigated the use of infrared and hall effect sensors integrated into a custom FAB for use in low-middle income countries. These studies developed new FAB devices with integrated sensors to capture their measurements. Although this is useful for individual research studies, this does not make use of current existing FAB devices and could be prohibitive due to cost restrictions or preferred choice of FAB type. A solution that could use existing prescribed FAB devices would make monitoring more accessible.

Low-cost accelerometers have shown potential for monitoring device wear in a range of applications [16,17,18]. They have an advantage over other sensors by being small, inexpensive, and able to directly measure the device’s movement. They could offer an alternative solution to previously described methods and would be affordable and simple to apply to existing FABs. However, for accelerometers to be used in this application, a robust wear algorithm needs to be developed.

This study presents a method for monitoring wear of FAB devices using a single 3-axis accelerometer and is a continuation of work by Silver et al. [19]. The method makes use of tailored open-source algorithms, creating a system that can be easily applied to all FAB devices. This new method was compared to self-reported wear to understand the agreement between the two measures and to validate the new method in an infant population.

## 2. Materials and Methods

### 2.1. Data Collection

Eleven families were recruited to participate in the study. Each of these families had an infant under the age of one already using an FAB and provided informed consent to participate in the study (approved by the local IRB No. 0029-21-SZMC). All infants had undergone a period of serial casting according to the Ponseti protocol and were using the FAB for a specific daily duration as advised by the managing orthopedic surgeon. Parents were asked to continue using the FAB as prescribed. Each infants’ FAB was fitted with a 3-axis accelerometer (activPAL PAL3-PAL Technologies, Glasgow, UK) with a sampling rate of 20 Hz. The device was adhered to the center of the FAB brace (Figure 1). As the data processing method uses a summation of the 3-axis acceleration signals, the device’s orientation was not considered important. The accelerometer was used to monitor the movement of the FAB device throughout the measurement period. Parents were also given a physical diary that was used to record the daily application and removal of the FAB in line with their treatment. Families were chosen for this study based on their highly compliant and reliable nature to provide a more “objective” diary report, against which the wear-time algorithm could be validated. At the end of the data collection, each participant removed the accelerometer from their FAB device and returned it to the research team, along with the diaries.

The raw accelerometer data were extracted using custom Python scripts (Python Software Foundation version 3.8.10), while the diaries were transcribed into a date-time format and saved for further analysis. The FAB wear algorithm’s development followed an open-access approach to ensure that the protocol was replicable and would enable comparison with future research. We used a proprietary algorithm designed to replicate the ActiGraph (ActiGraph LLC, Pensacola, FL, USA) activity count calculation from raw accelerometer data. An activity count is a way of quantifying the amount of movement within a given period and details on this method have been published previously by Brønd et al. [20]. The accelerometer data from the FAB monitors were used to calculate activity counts per second across the entire dataset. Following this, the activity count data was fed through an algorithm to determine periods when the device was worn and removed. The algorithm used to detect these periods was developed by Chadwell et al. [17] for the purpose of monitoring upper-limb prosthesis use. This algorithm was selected based on its application to assistive devices and the ability to alter its parameters to suit different applications. The algorithm evaluated wear and non-wear based on the activity count values within specific time periods, and each time period is assigned a classification based on an activity count cut-off threshold. This process is reassessed by investigating the duration of the classification and the duration of classifications preceding and processing the current time period. The activity count threshold and the duration for re-classifying time periods were adjusted based on an initial analysis of three participants’ data. All model parameters can be found in the Appendix A of this paper, and a detailed explanation of the algorithm is presented in the Supplementary Material of Chadwell et al. [17]. The algorithm provided wear and non-wear data on a second-by-second basis, and this was compared to the converted diary data to assess the agreement between the two measures.

### 2.2. Statistical Analysis

Agreement between the daily diary recordings and the algorithm measurements were performed using Bland Altman plots with limits of agreement. Daily wear time was calculated across all participants’ days. Any days with less than 20 h of data were removed from the analysis. This was done for both the algorithm and the diary measurements and was used to create the Bland Altman plots. To overcome the limitations of the Bland Altman plots, such as systematic errors within each day that cancel out when taking a daily average, agreement was also assessed hour by hour. All the participants’ data were separated out into time periods of one hour, which were then classified as worn or not worn. Any hours that contained both wear and non-wear were classified based on the longest time in each classification for that hour. This data was used to calculate Cohens Kappa coefficient across the dataset.

## 3. Results

The amount of data collected from each participant ranges from one to eight days, the large variation between participants being the result of early discontinued use of the FAB device. After removing any days with incomplete data, two participants were excluded from the analysis for having less than a full single day of data. When visually inspecting the algorithm and diary measured wear time with the raw accelerometer data, the measurements showed very similar agreement that corresponded with movement measured by the accelerometer, demonstrated in Figure 2.

Following classification, it was clear that on some days there were large differences (over five hours) between the reported wear time and the algorithm measured wear time. When visually assessing this data against the accelerometers’ vector magnitude signal, there were several periods where there was movement or lack of movement that was incorrectly reported in the diaries. Often these were near the end of that participant’s data collection, and it was assumed that the participant had failed to report these in the diary, thinking that the data collection period had ended (Figure 3a). Similarly, there were very long (>5 h) non-wear periods that were not reported in the middle of the data collection but were both very long (>5 h), and there was no movement in this period (Figure 3b). Finally, there were periods that were reported as non-wear, but there was clear, prolonged (>5 h) movement in the raw signal (Figure 3c). For these conditions, the data were removed from the dataset to ensure that issues in reporting did not affect the analysis.

A total of 45 days were used to create the Bland Altman plots (Figure 4). The Bland Altman plot quantifies the bias and the range of agreement within which 95% of the differences between self-reported and algorithm wear lie. The plot shows a −0.55 h in the diary measurements and limits of agreement ranging from −2.96 h to 1.96 h. Most of the differences between the reported and the measured wear time come from the differences in exact reporting times. However, there are a few outliers longer than two hours, resulting from the algorithm measuring daily wear more than what was reported. During these periods there is clear movement displayed from the accelerometer and they are either the result of miss-reporting or movement of the device that is not related to wear such as transportation.

To calculate the Cohens Kappa coefficient, any non-complete hours of data were removed from the dataset, which left 1218 h for analysis. Hourly classification was completed for both the diary data and the algorithm data and the Cohens Kappa coefficient for the entire dataset was 0.88, showing very high levels of agreement between the two measurements [21].

## 4. Discussion

This study is the first to use accelerometry to objectively measure FAB wear and validate this data against self-reported wear. The results show that the described method, which makes use of open-source algorithms, shows very high levels of agreement with diary recordings, both hourly and as average daily measurements. Furthermore, analysis of accelerometer data shows clear periods of movement missed by diary recordings, highlighting the advantage of using this objective method of measuring wear over self-reporting. The use of inexpensive accelerometers may enable FAB compliance monitoring to be commonplace in clinical practice, helping to better understand the impact of FAB use on idiopathic clubfoot treatment.

To our knowledge, this is the first study assessing the validity of an objective method of measuring FAB wear with self-reported diary measurements. Previous research on objective methods of monitoring FAB wear have mainly explored measures of compliance and used these measures to assess the validity of self-reporting. Morgenstein et al. [11] found that over three months of FAB use, self-reported compliance with treatment remained at 95% while objectively measured FAB compliance using temperature sensors dropped from 91.7% to 77.1%. This is important, as it shows that the difference between the measurements is not consistent, highlighting one of the issues with self-reporting. Although this is valuable information, the objective measurements presented by Morgenstein et al. [11] have not been previously validated, which makes it difficult to draw conclusions from this data. This study has validated our new method within a select infant population with parents that were known to be highly compliant with the FAB use and diary reporting. This validation enables future work that will aim to measure compliance in the wider population and understand reporting with unbiased family sampling.

The results presented in this work show that this new method has a high level of agreement with self-reported measures both hourly and daily. The Cohens Kappa analysis using hourly participant data shows the high level of agreement between the two measures within a single day. Meanwhile, the analysis of average daily wear using a Bland Altman plot shows that the devices had a good level of agreement per day, which is a much more likely use case for this data. There are some notable differences in the data when assessing the reported and measured wear boundaries. Although some of these differences are likely the result of donning and doffing the FAB device, many of the differences exceeded this expected time. This is likely the result of poor recall of when the device was used or the results of simplifying the diary data by rounding down the time. There were also several notable periods where the accelerometer recorded prolonged wear and non-wear periods that were misreported. This could be the result of non-wear related movements such as traveling with the device while it was not being worn. However, based on the variation in accelerometer counts and the raw vector magnitude data during these periods, it is more likely the result of misreporting. Given that families were selected based on their reliability to report diary wear and non-wear, and their FAB use compliance, this highlights the need for objective methods of monitoring FAB wear and emphasizes the benefits of this method over traditional self-reporting measures.

Compliance with FAB wear-time protocols reduce the likelihood of clubfoot recurrence and the need for future interventions such as re-casting and surgery. However, despite this, many parents are not compliant with these protocols and the need for future interventions is high [4,5]. Morgenstein et al. [11] showed that objectively measured compliance reduced by up to 15% over the course of 3 months. Similarly, Alves et al. [22] found that parental non-adherence to FAB use can affect 34% to 61% of children, which results in 5-to 17-fold higher odds of relapse. Factors associated with this lack of compliance are parent’s lower education and income level and demonstrating a lack of understanding of the importance of the bracing protocol [23]. This information needs to be highlighted to parents following the casting phase of the treatment. This research presents a monitoring method that can be easily applied to existing FAB devices with the aim of encouraging use of FAB monitoring in clinical practice. This could lead to increased FAB compliance through parents knowing that their adherence to the FAB wear-time protocol was being monitored [24,25] and reduce the likelihood of clubfoot recurrence.

The method for monitoring FAB use presented in this paper has an advantage over previously described objective methods by using low-cost and simple to apply accelerometers. The devices can be easily attached to any pre-existing FAB, preventing the need for new devices to be created or bought, which can limit use in communities where the cost of new devices is prohibitive. The method could enable large scale monitoring of FAB users, helping to better monitor FAB compliance and refine the FAB use guidance. Furthermore, open access algorithms make this method more easily reproducible and will encourage its use in future research and clinical practice. The software developed as part of this work has also been made available as open access and can be found in the Appendix A of this paper.

There are two key limitations of this study, the first being the quality of data that was used to validate this new objective method of monitoring FAB wear. This work used self-reporting as the criterium measure of wear, and although families were chosen based on their highly compliant and reliable nature, the differences observed between the algorithm- and diary-measured wear show that self-reporting is unreliable. Although, the results of this study show that the new objective method of monitoring wear performs better than self-reporting. To fully capture the validity of this method, future work should aim to validate this data against more robust criterium measures such as direct observations. The second limitation is the quantity of data used within the analysis. The dataset used for this analysis was compiled and assessed as one single dataset. To understand differences across participants and how each participant could impact the device’s measurements, more participants should be recruited for this research.

## 5. Conclusions

This work introduces and validates a novel objective method of monitoring FAB wear and non-wear using inexpensive and easy to apply accelerometers. The method shows substantial agreement with reported hourly and daily measurements, while offering more precision by detecting exact wear and non-wear changing times and wear periods missed by self-reporting. The method provides an advantage over existing objective monitoring devices as it can be easily applied to any existing FABs, preventing the need for bespoke monitoring devices. This method can facilitate increased research into FAB compliance and help enable FAB monitoring in clinical practice.

## Figures and Tables

**Figure 1 sensors-22-02433-f001:**
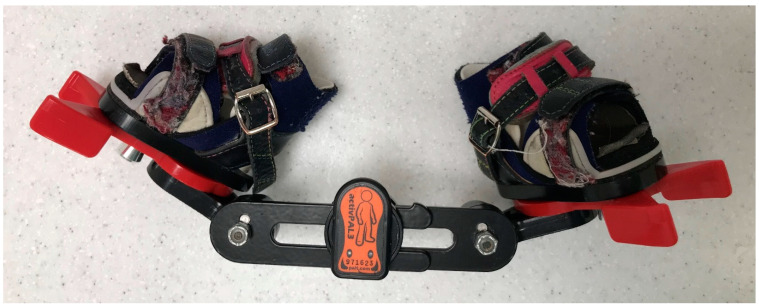
Data collection setup with activPAL PAL3 attached to a FAB.

**Figure 2 sensors-22-02433-f002:**
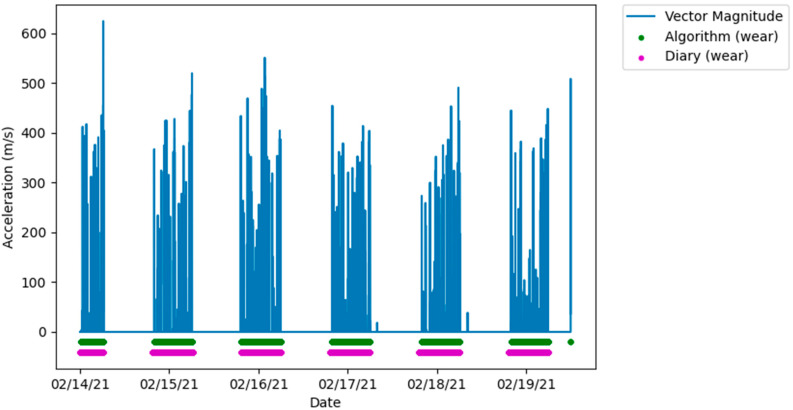
Plot showing a high level of agreement between algorithm and diary measured wear and non-wear, aligned with the vector magnitude from the raw accelerometer data.

**Figure 3 sensors-22-02433-f003:**
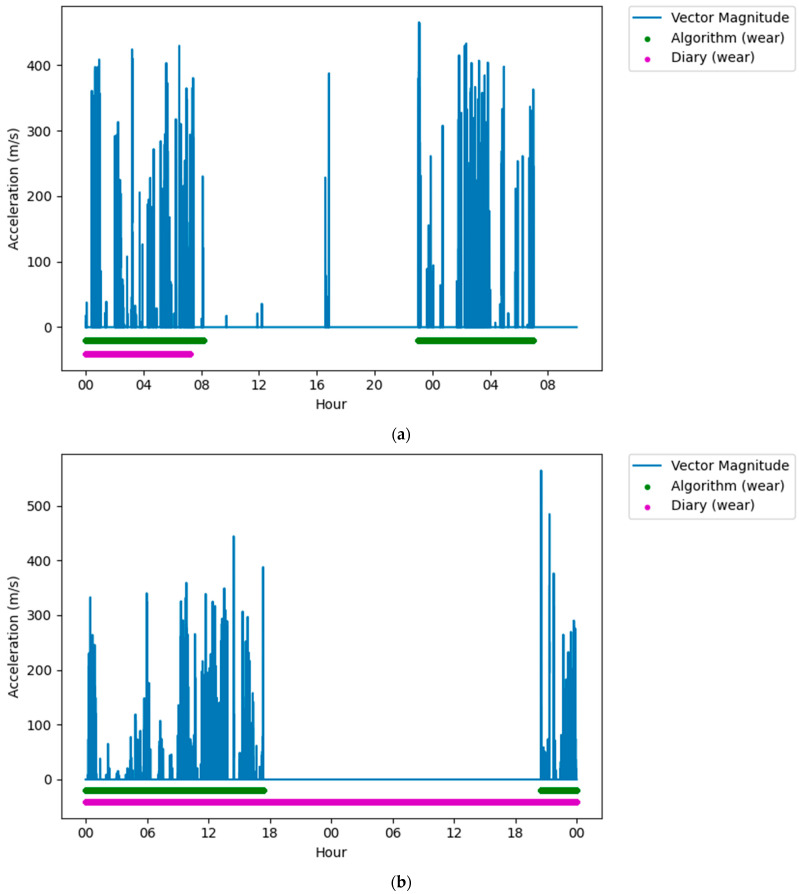
Figures showing issues with self-reported measures that were removed from the dataset. (**a**) Plot showing a long wear period not reported in the diary, but detected by the algorithm, near the end of the data collection period. (**b**) Plot showing a long non-wear period not reported in the diary, but detected by the algorithm, in the middle of a data collection period. (**c**) Plot showing a long wear period not reported in the diary, but detected by the algorithm, in the middle of a data collection period.

**Figure 4 sensors-22-02433-f004:**
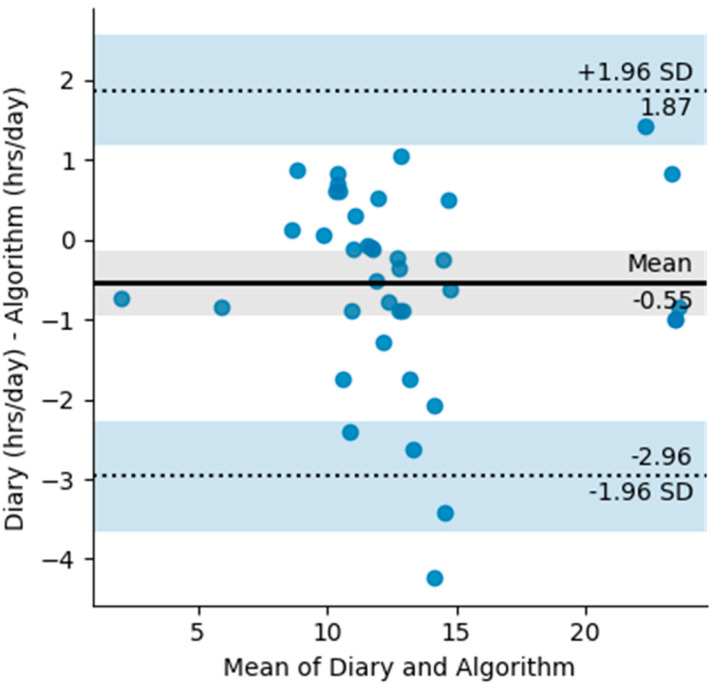
Bland Altman plot with limits of agreement for diary measured wear (hours/day) and algorithm measured wear (hours/day). The shaded areas are confidence interval limits for the mean and limits of agreement.

## Data Availability

The data are not publicly available due to privacy.

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
