# Peer review of "Measuring Foot Abduction Brace Wear Time Using a Single 3-Axis Accelerometer†"

_sensors, 2022, doi:10.3390/s22072433_

Round 1

Reviewer 1 Report

The authors presented their study on objective measurements of FAB wear time using accelerometer and validated the results with self-reporting by parents.

After reviewing the manuscript, the following questions were raised for me:

1)  The objective measurement with accelerometer was validated with self-reporting where, based on authors report, is not a reliable method (last paragraph, page 8). 

2) The data were fed to statistical analysis after eliminating the mismatched data between report and measurements, based on the authors interpretation. This would end to a good agreement of two methods and give a good validation for the suggested method. The authors tried to justify the mismatched results and remove them from analysis that obviously ended to agreement between two methods and validation of suggested method. I don't think the way that they interpreted the discrepant results is acceptable and they need to choose a reliable method to validate their measurement (Page 4, last sentence: For these conditions, where the measurements and diary are not match, the data were removed from the dataset to ensure that issues in reporting did not affect the analysis. Page 6, line 4: Most of the differences between the reported and the measured wear time come from the differences in exact reporting time. Page 6, line 6: ...either miss reporting or movement of the device that is not related to wear such as transportation. Page 7, line 1: To calculate the Cohens Kappa coefficient, any non-complete hours of data were removed from the dataset. 

3) Page 8, paragraph 2: This is mostly about the motivation of the study which I believe should not be addressed in the discussion section. 

Author Response

1)  The objective measurement with accelerometer was validated with self-reporting where, based on authors report, is not a reliable method (last paragraph, page 8).

This summary poorly explained the point being made and has been changed to better explain this. We are trying to say that although now this new method has been validated against self-reporting. To fully capture the true performance, further validation needs to take place against more robust data e.g. direct observations. Although, it's worth pointing out that these methods also come with drawbacks in the form of participant behaviour when being observed in controlled environments and the performance of direct observation itself. Therefore, this work provides very useful insight but this point should be considered when interpreting the data.

2) The data were fed to statistical analysis after eliminating the mismatched data between report and measurements, based on the authors interpretation. This would end to a good agreement of two methods and give a good validation for the suggested method. The authors tried to justify the mismatched results and remove them from analysis that obviously ended to agreement between two methods and validation of suggested method. I don't think the way that they interpreted the discrepant results is acceptable and they need to choose a reliable method to validate their measurement (Page 4, last sentence: For these conditions, where the measurements and diary are not match, the data were removed from the dataset to ensure that issues in reporting did not affect the analysis. Page 6, line 4: Most of the differences between the reported and the measured wear time come from the differences in exact reporting time. Page 6, line 6: ...either miss reporting or movement of the device that is not related to wear such as transportation. Page 7, line 1: To calculate the Cohens Kappa coefficient, any non-complete hours of data were removed from the dataset. 

There are 2 different aspects of your review that I would like to explain. Firstly, the following lines

  • Page 6, line 4: Most of the differences between the reported and the measured wear time come from the differences in exact reporting time
  • Page 6, line 6: ...either miss reporting or movement of the device that is not related to wear such as transportation.
  •  Page 7, line 1: To calculate the Cohens Kappa coefficient, any non-complete hours of data were removed from the dataset. 

The first point here is explaining where the majority of the differences between the measurements were coming from. This data was not removed from the dataset, this is from the data that was included in the analysis and is a point of discussion for the reader. The second point also comes from data that was not removed from the dataset and is included in the analysis. Again, it aims to offer an explanations as to why we were seeing longer duration discrepancies between the measurements and is highlighting where the 2 devices do not agree. The third point is explaining that non-complete hours were removed from the dataset to calculate cohens kappa. This most likely resulted in an extremely small percentage of data being removed from the dataset, at the end and beginning of each participants data collection, and was performed to ensure the cohens kappa coefficient was calculated correctly. Without doing this, small periods of time would be weighted equally to an entire hour and although would have had a negligible impact on the results, was still performed for correctness.

With regard to your interpretation of the data we removed from the dataset. The following is taken from the text and describes the data that was removed.

Following classification, it was clear that on some days there were large differences (over five hours) between the reported wear time and the algorithm measured wear time. When visually assessing this data against the accelerometers’ vector magnitude signal, there were several periods where there was movement or lack of movement that was incorrectly reported in the diaries. Often these were near the end of that participant’s data collection, and it was assumed that the participant had failed to report these in the diary, thinking that the data collection period had ended (Figure 3.a). Similarly, there were very long (>5hrs) non-wear periods that were not reported in the middle of the data collection but were both very long (>5hrs), and there was no movement in this period (Figure 3.b). Finally, there were periods that were reported as non-wear, but there was clear, prolonged (>5hrs) movement in the raw signal (Figure 3.c). For these conditions, the data were removed from the dataset to ensure that issues in reporting did not affect the analysis. Page 4

This describes a strict criteria where long (>5hours) periods of data did not agree with the algorithm output and on interpretation of the raw accelerometer data, showed clear signs of movement or lack of movement that was miss-reported. We discussed at length how to deal with this data as if it was included would significantly reduce the agreement between our results further showing that the new objective method performed better than self reporting (if you interpret as we did that the self reported data was incorrect based on the movement and lack of movement observed by the accelerometer). However, we chose to remove this data as it was clear that these long periods were issues in remembering to report rather than differences in reporting performance. This was confirmed anecdotally with a few participants that said they forgot to report some periods of time. We hoped that by removing the data we are highlighting that remembering to report is clearly an issue that has been explored in other work and instead shows the performance of our method against self-reporting. The results show they generally have good agreement but there are periods that they don't agree and these are discussed.

3) Page 8, paragraph 2: This is mostly about the motivation of the study which I believe should not be addressed in the discussion section.

Although this paragraph does discuss motivation for the study it also puts the outcomes of the work into context and discusses how this new method is a positive move in making FAB monitoring common place in clinical practice. Some sentences have been changed to better explain this.

Hopefully, these explanations address your comments and we would be happy to receive further feedback.

Reviewer 2 Report

The authors validate the functionality of the 3-axis accelerator to monitor the FAB.

Although the author provided the algorithm link (github) and it is presented in [17]. The algorithm should be briefly explained in the paper.

Author Response

Thanks for your review. In regard to your comment below, I have added in a short description of the algorithms method. This summarises the way the algorithm performed it's classification at a high level. We believe that going into further depth would make this section confusing and hope that this provides enough detail that the reader could explore the previous paper if interested. 

Although the author provided the algorithm link (github) and it is presented in [17]. The algorithm should be briefly explained in the paper.

Round 2

Reviewer 1 Report

I would like to thank authors for their response, clarification, and edition. In version two, one sentence in page 3, paragraph 2, has been prolonged and now it is so awkward. The three-and-half-line sentence would be clearer and more appropriate if it is broken to at least two separate ones.   

Author Response

I have split the sentence up to make it more clear.

Thanks